# Effects of Different Proportions of *Amaranthus hypochondriacus* Stem and Leaf Powder Inclusions on Growth Performance, Carcass Traits, and Blood Biochemical Parameters of Broilers

**DOI:** 10.3390/ani13182818

**Published:** 2023-09-05

**Authors:** Ying Ren, Lingyu Liu, Shilong Zhou, Yantao Li, Yan Wang, Kang Yang, Wenxun Chen, Shengjun Zhao

**Affiliations:** Hubei Key Laboratory of Animal Nutrition and Feed Science, Wuhan Polytechnic University, Wuhan 430023, China; renying@whpu.edu.cn (Y.R.); leanliu1998@163.com (L.L.); zhousl0514@163.com (S.Z.); lyt906040062@163.com (Y.L.); w001003y@163.com (Y.W.); 18608554499@163.com (K.Y.); cwx17@whpu.edu.cn (W.C.)

**Keywords:** *Amaranthus hypochondriacus*, growth performance, apparent nutrient digestibility, carcass traits, blood biochemical, broiler

## Abstract

**Simple Summary:**

Considering the shortage of conventional feeds, such as corn and soybeans, researchers have gradually started to develop and apply unconventional feed resources. As an unconventional feed resource, *Amaranthus hypochondriacus* is a potential high-quality feed resource. Hence, this study applied different proportions of *Amaranthus hypochondriacus* stem and leaf powder in experimental diets to assess its nutritional value for broilers.

**Abstract:**

This experiment aimed to study the effects of different proportions of *Amaranthus hypochondriacus* stem and leaf powder (AHSL) on the growth performance, apparent nutrient digestibility, carcass traits, meat quality, and blood biochemical parameters of broilers from day 1 to day 42. The experiment utilized a single-factor experimental design, with a total of 216 one-day-old male broilers (Ross 308) randomly assigned to three dietary treatment groups (eight replicate cages of nine birds per cage). The dietary treatments included a control diet, a 3% AHSL diet and a 6% AHSL diet for days 0–21. Then, the 3% and 6% AHSL diets were changed to 5% and 10% AHSL for days 22–42. The results showed that the inclusion levels of AHSL did not affect growth performance, carcass traits, or meat quality on days 21 and 42 (*p* > 0.05). However, the inclusion levels of AHSL decreased the apparent nutrient digestibility (AND) of dry matter (DM) (*p* ˂ 0.001) and neutral detergent fiber (NDF) (*p* ˂ 0.001) and increased the serum concentration of phosphorus (*p* ˂ 0.001) on day 21. On day 42, the inclusion levels of AHSL decreased the AND of DM (*p* = 0.025) and NDF content (*p* ˂ 0.001), but increased the AND of crude protein (CP) (*p* = 0.004). In particular, the diet containing 10% AHSL significantly increased the serum enzyme activity of alkaline phosphatase (ALP) (*p* = 0.046) and the serum concentration of total protein (TP) (*p* ˂ 0.001) on day 42. Overall, AHSL can be used as a new and effective feed ingredient in broiler diets. It can replace part of the corn–soybean meal diet without any adverse effects, which is beneficial for conserving feed resources. Additionally, AHSL can be included at a level of up to 10% during the broiler growth period.

## 1. Introduction

Feed is the largest expense in broiler production, accounting for approximately 60–80% of the total cost [1]. Corn and soybean are the primary sources of energy and protein in broiler feed [2]. Considering the shortage of corn and soybean, grain amaranth may be used as a new feed resource to replace a portion of corn or soybean meal in broiler diets [3]. *Amaranthus* belongs to the Amaranthaceae family and is a fast-growing, high-yielding grain and forage crop that is rich in nutrients. It originates from the tropical and subtropical regions of Central America and Southeast Asia [4,5]. Currently, there are three main species of *Amaranthus* that are grown worldwide as cereals, including *Amaranthus cruentus*, *Amaranthus caudatus*, and *Amaranthus hypochondriacus* [6]. *Amaranthus hypochondriacus* is a C4 plant that exhibits salt tolerance, strong stress resistance, and stress tolerance [7]. It has a high nutritional value and is roughly equivalent to corn in terms of gross energy [8]. Amaranth oil has also been found to contain squalene, which is an important precursor of all steroids [9]. The squalene, fibers, and tocotrienols present in amaranth grains have been proven to reduce blood serum cholesterol levels [10,11]. Furthermore, the stems and leaves of *Amaranthus hypochondriacus* have a very high crude protein content, ranging from 16 to 23 percent under dry matter conditions. They also contain lysine, methionine, and other important amino acids. Its lysine concentration is particularly high at around 1% [12]. 

Previous research has revealed that adding untreated and heat-treated grain amaranth meals to broiler diets as a replacement for animal protein (meat and bone meal) has no discernible impact on the growth performance of broilers [13]. Fasuyi et al. [14] reported that sun-dried *Amaranthus cruentus* leaf meal may serve as a nutrient-rich feed source for broilers and can be included in their diets. When included at a proportion of 20% in broilers’ diets, it does not affect the health of broilers. Menyelo et al. [15] found no significant differences in the feed intake, body weight, or feed efficiency of broilers when fed diets containing 0%, 5%, 10%, 15%, and 20% *Amaranthus cruentus* leaf meal. However, there is limited literature on the application of *Amaranthus hypochondriacus* in broiler nutrition. Therefore, this study aimed to test different proportions of *Amaranthus hypochondriacus* stem and leaf powder (AHSL) in experimental diets to assess the nutritional value for broilers in terms of their growth performance, apparent nutrient digestibility, carcass traits, meat quality, and blood biochemical parameters.

## 2. Materials and Methods

### 2.1. Animal Care

The experiment was conducted in the Poultry Nutrition and Metabolism Lab of Wuhan Polytechnic University. The animal handling and management procedures were reviewed and approved by the Ethics and Research Committee of Wuhan Polytechnic University (Hubei, China; approval number: 2010-0029).

### 2.2. Preparation of Amaranth

Before formulating the diets, the stem and leaf powder of *Amaranthus hypochondriacus* (Fuxian Agricultural Technology Co., Ltd., Xiaogan, China) was pulverized. The powder was then analyzed for crude protein (CP), ether extract (EE), crude fiber (CF), calcium (Ca), and phosphorous (P), following the standard procedures of the Association of Official Analytical Chemists (AOAC) [16]. The chemical composition of *Amaranthus hypochondriacus* stem and leaf powder is presented in Table 1. Subsequently, the digestible energy and metabolizable energy values of the ingredients were calculated using the formula recommended by the NRC (2004). The experimental diets were formulated based on the analyzed values. 

### 2.3. Animals and Diets

The experiment utilized a single-factor experimental design. A total of 216 one-day-old male broilers (Ross 308) were obtained from a local hatchery (Zhengda Agriculture and Animal Husbandry Food Co., Ltd., Xiangyang, China). These broilers were randomly assigned to three dietary treatment groups (8 replicate cages of 9 birds per cage).

Table 2 presents the ingredient composition of the experimental diets. The dietary treatments included a control diet, a 3% AHSL diet and a 6% AHSL diet for 0–21 d. Then, the 3% and 6% AHSL diets were changed to 5% and 10% AHSL diets, respectively, for days 22–42, and the diets were provided in powdered form. According to the guidelines made by the NRC (2004), all the experimental diets were designed to meet or exceed the nutritional requirements for broiler chicks at the ages of 21 and 42 days, and the diets were isocaloric and isonitrogenous. Additionally, all the diets had similar nutrient compositions.

### 2.4. Rearing Management

Prior to the arrival of the broilers, the experimental house, feed, and water troughs were thoroughly cleaned and disinfected. The broilers were housed in steel floor cages within a climate-controlled environment with constant lighting, and they were given free access to feed and water. The room temperature was maintained at 35 °C during the first week and gradually decreased by 3 °C per week until it reached 23 °C. Until day 3, the lighting schedule consisted of 23 h of light and 1 h of darkness, followed by 18 h of light and 6 h darkness. The relative humidity in the birdhouse was set at 55%. During the first week, all the chicks were administered a broiler coccidiosis vaccine through drinking water. Additionally, they were inoculated against the Newcastle disease virus at around 7 and 14 days of age.

### 2.5. Growth Performance

The body weight of the broilers was recorded on d 1. After a weekly 4-h fast, feed intake and body weight were recorded for each repetition (cage). Body weight gain (BWG), the average daily feed intake (ADFI), and the feed conversion ratio (FCR) were calculated based on weekly data recorded and analyzed for the following time periods: 1–21 days and 22–42 days.
(1)$$Feed\;conversion\;ratio=\frac{Feed\;intake\;(g)}{Body\;weight\;gain\;(g)}$$

### 2.6. Apparent Nutrient Digestibility

Fresh fecal samples were collected on days 19–21 and 40–42 and were stored at −20 °C for testing. The feed and fecal samples were dried in a forced air oven at 65 °C for 2 days. Then, the samples were left to air-dry for 1 day in order to obtain dried samples. The samples were pulverized using a grinder and were passed through a 1 mm sieve. Titanium dioxide (TiO_2_) was used as an exogenous indicator to determine the apparent nutrient digestibility (AND). The TiO_2_ content was measured following the method described by Short [17]. The dry matter (DM), crude protein (CP), and ether extract (EE) contents were measured using the methods outlined by the Association of Official Analytical Chemists (AOAC) [16]. The neutral detergent fiber (NDF) content was measured according to the method developed by Van Soest [18]. The formula that was used for the calculation is as follows:(2)$$AND(\%)=1-\left\{\left(\frac{TiO_{2}\;\%\;diet}{TiO_{2}\;\%\;fecal}\right)\times \left(\frac{Nutrient\;\%\;fecal}{Nutrient\;\%\;diet}\right)\right\}$$

### 2.7. Blood Biochemical Parameters

On d 21 and 42, two broilers from each replicate were randomly selected for blood sampling from the wing vein. Serum was obtained by centrifugation at 3000 r/min and 4 °C for 10 min. Then, the serum was stored at −20 °C for determination of the blood biochemical parameters.

The biochemical parameters of serum, including the enzymatic activities of aspartate aminotransferase (AST), alkaline phosphatase (ALP), and lactate dehydrogenase (LDH), as well as the concentrations of total bilirubin (TB), direct bilirubin (DB), total protein (TP), albumin (ALB), total cholesterol (TC), triglyceride(TG), glucose (GLU), calcium (Ca), phosphorus (P), high density lipoprotein (HDL), uric acid (UA), gamma-glutamyltransferase (GGT), and creatine kinase (CK), were determined using an automatic biochemistry analyzer (7100, Hitachi, Tokyo, Japan) [19].

### 2.8. Carcass Traits and Meat Quality

On d 42, after 4 h of fasting and based on the average body weight, sixteen broilers were randomly selected from each treatment (two broilers per replication), weighed, and then slaughtered. The carcasses were carefully dissected, and the samples, including the carcass, breast, thigh, abdominal fat, liver, thymus, spleen, pancreas, and bursa of Fabricius, were collected for further analysis. The relative weights of the organs were calculated as a percentage of the live body weight.

The individual breast and thigh samples were then placed in valve bags and were stored at −20 °C for later meat quality analysis. The pH values were measured separately at different parts using a pH meter (STARTER2100, Shanghai, China). A colorimeter (CR-410, Konica Minolta, Tokyo, Japan) was used to determine the Lightness (L*), Redness (a*), and Yellowness (b*) of the color of the meat. Three measurements were taken for each meat sample, and the average value was recorded. Cooking loss was estimated by heating the sample at 98 °C for 20 min, allowing it to cool to room temperature, and then weighing it. The percentage of cooking loss was calculated using Equation (3), where W_precook_ and W_cook_ represent the weights before and after cooking, respectively. Following the steps outlined by Papinaho and Fletcher [20], the cooked muscle was cut into a 3 cm × 1 cm × 1 cm shape perpendicular to the muscle fibers, and the shear force of the muscle was measured using a muscle tenderness meter (TA500 Lloyd Texture Analyzer fitted with a triangular Warner–Bratzler shear, Lloyd instruments, Bognor Regis, UK). The inosine monophosphate content in the breast and thigh samples was determined using the HPLC method described by Zhang et al. [21]. Total cholesterol was measured using specific assay kits (Nanjing Jiancheng Bioengineering Institute, Nanjing, China) [22].
(3)$$Cooking\;loss=\frac{W_{percook}-W_{cooked}}{W_{percook}}\times 100\%$$

### 2.9. Statistical Analysis

The experimental data were analyzed using a one-way ANOVA with IBM SPSS Statistical 26.0 (SPSS Inc., Chicago, IL, USA). Duncan’s multiple range tests were used to determine statistically significant differences, with a significance level set at *p* < 0.05. The results are presented as means with the standard error of the means (SEM).

## 3. Results

### 3.1. Growth Performance

The results for BWG, ADFI, and FCR on d 21 and d 42 of the broilers are summarized in Table 3. None of the indices showed statistical significance (*p* > 0.05) on both d 21 and d 42.

### 3.2. Apparent Nutrient Digestibility

The inclusion of 3% and 6% AHSL in the diet significantly decreased the AND of DM and NDF on d 21 (*p* < 0.001). However, AHSL inclusion levels had no effect on the AND of CP and EE on d 21 (*p* > 0.05). On d 42, the AND of DM and NDF in the diet with 5% and 10% AHSL inclusion levels decreased significantly compared to the control diet (*p* = 0.025; *p* < 0.001), whereas the AND of CP increased significantly (*p* = 0.004). The AHSL inclusion levels had no effect on the AND of EE on d 42 (*p* > 0.05) (Table 4).

### 3.3. Blood Biochemical Parameters

The serum concentration of phosphorus in the diets with AHSL inclusion levels of 3% and 6% was significantly higher than that in the control diet (*p* < 0.001) on d 42. The serum enzyme activity of ALP and the serum concentration of TP in the diet with an AHSL inclusion level of 10% increased significantly (*p* = 0.046; *p* < 0.001) compared to the control diet on d 42. AHSL inclusion levels showed no significant differences (*p* > 0.05) in their effects on the other blood biochemical parameters on d 21 and 42 (Table 5).

### 3.4. Carcass Traits and Meat Quality

There were no statistically significant differences (*p* > 0.05) in the carcass traits between the different experimental groups (Table 6). However, the results showed that the values for all the carcass traits were higher for the broilers consuming diets with 10% AHSL inclusion levels compared to broilers consuming the control diet.

The meat quality did not show statistical significance (*p* > 0.05) in both the breast and thigh (Table 7). However, the pH and redness of the breast, as well as the lightness and redness of the thigh, were higher for the broilers consuming diets with 5% and 10% AHSL inclusion levels compared to broilers consuming the control diet. The lightness of the breast and the inosine monophosphate content of the thigh was lower than that observed for the broilers consuming the control diet. 

## 4. Discussion

### 4.1. Growth Performance

This study found that AHSL inclusion levels did not affect the BWG, ADFI, and FCR of broilers at 21 days and 42 days of age. These results are consistent with previous studies in broilers. Waldroup et al. [23] found that broiler chickens can consume *Amaranthus hypochondriacus* and *Amaranthus cruentus* grain, and up to 20% of raw or autoclaved amaranth can be added to their diets without significantly affecting BWG, AFI, and FCR. Similar findings were reported by Rouckova et al. [24] and Manyelo et al. [25], who fed broilers with various amaranth incorporations and observed no effect on broiler growth performance, consistent with Waldroup’s results. These studies indicate that amaranth can be used as feed for broilers without affecting their growth performance. However, Tillman et al. [26] suggested that *Amaranthus cruentus* can be included in broiler diets at a maximum level of 40%, but only after undergoing appropriate processing, such as extrusion or autoclaving. Ravindran et al. [27] conducted a study in which raw *Amaranthus hypochondriacus* was added to broiler diets at levels of 0%, 20%, 40%, and 60%, replacing corn, soybean meal, and meat meal. They compared feed intake and energy utilization values and found that increasing the content of raw *Amaranthus hypochondriacus* in diets inhibited the AFI and BWG of broilers. The existence of anti-nutrient factors, such as oxalic acid and phytic acid, in raw *Amaranthus hypochondriacus* limits its utilization in livestock and poultry production. Tillman et al. [26] and Ravindran et al. [27] eliminated these anti-nutrient factors through heat treatment, autoclaving, and other appropriate ways, so that the inclusion levels of *Amaranthus hypochondriacus* reached 60% [28]. In contrast, Ghamsari et al. [29] showed that dietary inclusion with 4% or 6% amaranth decreased the body weight, ADG, and FCR of broilers. In this experiment, the species and processing methods of amaranth may be different from those in Ghamsari’s experiment, so the test results are different.

### 4.2. Apparent Nutrient Digestibility

The maintenance of good animal performance is closely related to the digestion and metabolism of each nutrient in the body. Increased nutrient digestibility can promote the absorption of nutrients, such as dry matter, nitrogen, and gross energy, thereby promoting growth performance to some extent. This study found that the inclusion levels of AHSL decreased the AND of dry matter and neutral detergent fiber on d 21 and 42, but increased the AND of crude protein on d 42. These findings are consistent with a study conducted by Xia et al. [8], who reported that the addition of 10% grain amaranth reduced the digestibility of dry matter in pregnant sows. Manyelo et al. [25] also reported that the digestibility of crude fiber, crude protein, and ash in Ross 308 broilers consuming a diet containing 5% amaranth leaf meal was higher than in broilers consuming a diet containing 0%, 10%, 15%, and 20% amaranth leaf meal. The decreased digestibility of dry matter and neutral detergent fiber due to the inclusion of AHSL in the diet may be due to the higher lignin content in the stem and leaf meal, which inhibits the digestibility of these components [30]. In addition, broilers mainly rely on cecum microbial fermentation to degrade fiber, which has certain limitations on fiber degradation. When broilers are fed high-fiber diets, they need longer adaptation times [31]. The reduced digestibility of fiber in a high-fiber diet leads to the enrichment of fiber in the gut, which, in turn, reduces the digestibility of dry matter. These reasons may explain the results of this experiment.

### 4.3. Blood Biochemical Parameters

Serum biochemical indexes can most directly and effectively reflect the nutrient deposition status, health status, metabolic status, and breeding status of livestock and poultry. Serum TP content reflects the protein metabolism capacity and immune function of the body, whereas ALP reflects the growth and development of endoskeleton and the deposition of calcium and phosphorus. This study found that the inclusion levels of AHSL significantly increased the serum enzyme activity of ALP and TP concentration on d 42. Rouckova et al. [24] reported that amaranth grain diets increased the plasma TP of broilers. However, Popiela et al. [32] and Alizadeh et al. [29] reported that serum TP content and the activity of ALP were not significantly affected by dietary treatments. These studies indicate that amaranth may increase TP content in the blood of broilers, which may be attributed to the high amount of 20-hydroxyecdyone present in the diet. It has been reported that 20-hydroxyecdyone, which is a phytochemical in amaranth, can regulate protein synthesis [33], but further study is needed to understand its specific mechanism. Except for phosphorus, TP, and ALP, the majority of the serum biochemical indices evaluated in this study were not significantly affected by dietary changes. Previous studies conducted by Alizadeh et al. [29], Kroliczewska et al. [34], and Longato et al. [35] have reported that diets containing amaranth granules reduced serum cholesterol levels in broilers and laying hens. This is consistent with the claim that squalene, which is present in amaranth, can lower serum cholesterol. The HMG-CoA reductase enzyme is a target for hypercholesterolemia; however, amaranth can be used as a powerful inhibitor of the HMG-COA reductase enzyme to reduce cholesterol by inhibiting or reducing its activity. The results of this experiment demonstrate that AHSL could reduce the total cholesterol in serum, but this difference was not significant. The reason for this may be that only the stems and leaves of *Amaranthus hypochondriacus* were used in this study, rather than the whole plant, which may affect the content of squalene in *Amaranthus hypochondriacus*. This study indicated that including AHSL in the diet had no adverse effects on the serum biochemical indices of broilers.

### 4.4. Carcass Traits and Meat Quality

Slaughter performance is an important indicator used to measure the meat production capacity of broilers, as it is influenced by factors such as nutrient levels, environmental conditions, and diseases. The meat quality indices measured in this study included pH, meat color, shear force, and cooked meat rate, which are all important indicators used to assess muscle quality, including tenderness and flavor. According to Fasuyi et al. [36], Mbugua et al. [37], and Pisarikova et al. [13], broiler diets could include amaranth as a nutrient-rich source at an inclusion level of 40% without significant differences in the carcass traits and meat quality of broilers. This study found that AHSL inclusion levels did not affect the carcass traits and meat quality on d 21 and 42. This is consistent with our research results. In contrast, Manyelo et al. [15] found that broilers fed a diet with 20% amaranth leaf meal (ALM) had higher drumstick weights and abdominal fat compared to broilers fed a diet with 5%, 10%, and 15% ALM. Additionally, the brightness (L*) of the breast meat improved with inclusion levels of 5%, 10%, and 15% ALM, but the other carcass traits and meat color indices were not significantly affected by ALM inclusion levels. As the amaranth levels increased, there was a corresponding increase in the chlorophyll content and the carotene levels in the diet, which affected the brightness of the muscles. The maximum inclusion level of AHSL in this study was 10%, which is lower than the inclusion levels in the literature mentioned above. This may explain the results of this experiment.

## 5. Conclusions

AHSL can be used as a new and effective feed ingredient in broiler diets, replacing a portion of the corn–soybean meal diet, with no adverse effects on growth performance, apparent nutrient digestibility, blood biochemical parameters, carcass traits, and the meat quality of broilers, which helps to conserve feed resources. Additionally, during the growth period of broilers, the inclusion level of AHSL can reach up to 10%. 

There is great research potential for further exploring the application of *Amaranthus hypochondriacus* as an available protein resource feed material in broiler production. In the future, further studies can explore the appropriate supplemental levels of *Amaranthus hypochondriacus* resources using different processing technologies in broiler production. Additionally, investigating the potential of *Amaranthus hypochondriacus* in different animal production contexts is also worth exploring.

## Figures and Tables

**Table 1 animals-13-02818-t001:** The chemical composition of *Amaranthus hypochondriacus* stem and leaf powder.

Item (%)	Value
Digestible energy (MJ/kg)	8.41
Metabolizable energy (MJ/kg)	3.52
Crude protein	16.19
Ether extract	1.70
Crude fiber	34.20
Calcium	2.19
Phosphorus	0.07

**Table 2 animals-13-02818-t002:** Ingredient and nutrient compositions of experimental diets on d 21 and 42.

Treatment	0–21 Days			22–42 Days		
	Control	3% AHSL	6% AHSL	Control	5% AHSL	10% AHSL
Ingredient composition (%)						
Corn	55.60	52.17	48.55	59.88	54.10	48.20
Soybean meal	38.57	38.08	37.61	33.38	32.56	31.78
AHSL ^1^	0.00	3.00	6.00	0.00	5.00	10.00
Soybean oil	2.61	3.70	4.95	3.50	5.38	7.37
Limestone powder	1.33	1.12	0.92	1.45	1.12	0.75
Dicalcium phosphate	1.30	1.34	1.38	1.05	1.10	1.20
Salt	0.35	0.35	0.35	0.35	0.35	0.35
DL-Methionine	0.20	0.20	0.20	0.15	0.15	0.15
Vitamin premix ^2^	0.04	0.04	0.04	0.04	0.04	0.04
Mineral premix ^3^	0.00	0.00	0.00	0.20	0.20	0.20
Titanium dioxide	0.40	0.40	0.40	0.40	0.40	0.40
Calculated energy and nutrient composition (% unless stated otherwise)						
Metabolizable energy (MJ/kg)	12.20	12.17	12.18	12.58	12.55	12.55
Crude protein	21.51	21.51	21.51	19.51	19.51	19.51
Ether extract	5.14	6.13	7.27	6.09	7.79	9.61
Crude fiber	2.16	2.69	3.22	2.06	3.65	5.24
Lysine	1.29	1.26	1.24	1.14	1.10	1.07
Methionine	0.56	0.55	0.54	0.48	0.47	0.45
Methionine + Cysteine	0.85	0.85	0.82	0.74	0.71	0.69
Tryptophan	0.29	0.28	0.28	0.26	0.25	0.24
Threonine	0.88	0.86	0.84	0.80	0.77	0.73
Calcium	1.00	1.00	1.00	0.95	0.96	0.96
Available phosphorus	0.69	0.68	0.68	0.61	0.60	0.61

Note: ^1^ AHSL: *Amaranthus hypochondriacus* stem and leaf powder; ^2^ the vitamin premix provided the following per kilogram of diet: Vitamin A 12500 IU, Vitamin D3 2500 IU, Vitamin E 30 IU, Vitamin K3 2.65 mg, Vitamin B12 0.025 mg, Vitamin B2 6 mg, Vitamin B12 0.025 mg, Biotin 0.0325 mg, Folic acid 1.25 mg, and Niacin 12 mg; ^3^ the mineral premix provided the following per kilogram of diet: Cu 8 mg, Zn 75 mg, Fe 80 mg, Mn 100 mg, Se 0.30 mg, and I 0.35 mg.

**Table 3 animals-13-02818-t003:** Effect of feeding *Amaranthus hypochondriacus* stem and leaf powder on growth performance.

Items ^1^	0–21 Days					22–42 Days				
	Control	3% AHSL	6% AHSL	SEM ^2^	*p* Value	Control	5% AHSL	10% AHSL	SEM ^2^	*p* Value
Body weight (g)	541.94	579.78	571.76	7.065	0.064	1709.63	1735.61	1671.07	25.617	0.606
BWG (g/d)	23.72	25.54	25.15	0.336	0.062	55.36	54.94	52.39	1.079	0.497
ADFI (g/d)	32.26	34.19	34.16	0.502	0.203	102.55	106.75	103.93	1.919	0.680
FCR (g/g)	1.363	1.341	1.359	0.016	0.859	1.866	1.944	1.986	0.030	0.270

The data are the mean values obtained from eight replicates per treatment (nine chickens per replicate). ^1^ Body weight, broiler weight at slaughter; BWG, body weight gain; ADFI, average daily feed intake; and FCR, feed conversion ratio. ^2^ SEM: standard error of the means.

**Table 4 animals-13-02818-t004:** Effect of feeding *Amaranthus hypochondriacus* stem and leaf powder on apparent nutrient digestibility.

Items ^1^ (%)	0–21 Days					22–42 Days				
	Control	3% AHSL	6% AHSL	SEM ^2^	*p* Value	Control	5% AHSL	10% AHSL	SEM ^2^	*p* Value
DM	72.91 ^a^	71.50 ^b^	71.51 ^b^	0.140	˂0.001	72.73 ^a^	71.87 ^b^	71.96 ^b^	0.148	0.025
CP	57.62	58.32	60.39	1.288	0.700	60.69 ^b^	69.52 ^a^	65.43 ^a^	1.293	0.004
EE	87.43	88.66	86.61	0.519	0.289	78.90	80.96	83.64	0.919	0.095
NDF	41.56 ^a^	30.23 ^b^	27.52 ^b^	2.003	˂0.001	43.23 ^a^	23.95 ^b^	26.78 ^b^	2.684	˂0.001

The data are the mean values obtained from eight replicates per treatment (nine chickens per replicate). ^1^ DM, dry matter; CP, crude protein; EE, ether extract; NDF, neutral detergent fiber. ^2^ SEM: standard error of the means. ^a,b^ The means with no common superscripts within each row are significantly different (*p* < 0.05).

**Table 5 animals-13-02818-t005:** Effect of feeding *Amaranthus hypochondriacus* stem and leaf powder on blood biochemical parameters.

Items ^1^	0–21 Days					22–42 Days				
	Control	3% AHSL	6% AHSL	SEM ^2^	*p* Value	Control	5% AHSL	10% AHSL	SEM ^2^	*p* Value
TB (g/L)	0.52	0.55	0.53	0.017	0.840	0.43	0.38	0.44	0.024	0.568
DB (g/L)	0.06	0.08	0.07	0.004	0.203	0.08	0.07	0.08	0.004	0.205
TP (g/L)	2.21	2.40	2.23	0.052	0.276	1.57 ^b^	1.66 ^b^	2.41 ^a^	0.097	˂0.001
ALB (g/L)	1.24	1.29	1.22	0.014	0.102	1.24	1.37	1.25	0.026	0.063
AST (U/L)	199.67	196.43	191.50	3.787	0.695	235.50	263.50	250.29	6.992	0.297
ALP (U/L)	6631.75	7992.83	6896.67	385.299	0.330	2826.50 ^ab^	2314.67 ^b^	3985.71 ^a^	297.315	0.046
TC (nmol/L)	146.63	134.69	136.40	3.500	0.374	122.01	126.55	119.77	2.754	0.617
TG (nmol/L)	22.14	18.25	18.52	1.096	0.316	32.10	23.79	30.03	3.134	0.572
Glu (nmol/L)	192.82	195.23	192.20	2.935	0.912	203.50	197.83	213.14	5.161	0.487
Ca (nmol/L)	11.29	10.95	11.20	0.171	0.743	11.66	10.29	11.39	0.294	0.140
P (nmol/L)	5.51 ^b^	7.01 ^a^	6.85 ^a^	0.187	˂0.001	6.48	6.27	6.86	0.187	0.439
HDL (nmol/L)	136.29	127.01	124.63	3.363	0.369	101.69	114.70	104.56	2.620	0.110
UA (nmol/L)	4.21	4.07	4.09	0.375	0.989	6.14	5.46	3.54	0.495	0.067
GGT (U/L)	14.33	13.43	16.00	0.595	0.182	15.50	17.67	15.71	0.812	0.526
CK (U/L)	2044.67	1882.29	2014.75	117.836	0.854	5719.83	4595.33	6475.71	508.934	0.330
LDH (U/L)	1012.90	772.17	732.43	57.071	0.108	998.80	969.81	893.04	66.266	0.809

The data are the mean values obtained from eight replicates per treatment. ^1^ TB, total bilirubin; DB, direct bilirubin; TP, total protein; ALB, albumin; AST, aspartate aminotransferase; ALP, alkaline phosphatase; TC, total cholesterol; TG, triglyceride; GLU, glucose; Ca, calcium; P, phosphorus; HDL, high density lipoprotein; UA, uric acid; GGT, gamma-glutamyltransferase; CK, creatine kinase; and LDH, lactate dehydrogenase. ^2^ SEM: standard error of the means. ^a,b^ The means with no common superscripts within each row are significantly different (*p* < 0.05).

**Table 6 animals-13-02818-t006:** Effect of feeding *Amaranthus hypochondriacus* stem and leaf powder on carcass traits.

Items (%)	Control	5% AHSL	10% AHSL	SEM ^1^	*p* Value
Carcass	70.48	70.76	71.07	0.416	0.855
Breast	17.66	17.61	17.72	0.397	0.995
Thigh	15.02	15.68	15.22	0.259	0.584
Abdominal fat	1.08	1.09	1.28	0.086	0.562
Liver	1.72	1.87	1.77	0.066	0.665
Thymus	0.11	0.10	0.12	0.009	0.748
Spleen	0.10	0.11	0.10	0.011	0.950
Pancreas	0.17	0.18	0.19	0.007	0.439
Bursa of Fabricius	0.15	0.16	0.17	0.010	0.689

The data are the mean values obtained from eight replicates per treatment (two chickens per replicate). ^1^ SEM: standard error of the means.

**Table 7 animals-13-02818-t007:** Effect of feeding *Amaranthus hypochondriacus* stem and leaf powder on meat quality.

Items	Control	5% AHSL	10% AHSL	SEM ^1^	*p* Value
**Breast**					
pH	6.26	6.50	6.53	0.066	0.185
Lightness (L*)	89.25	87.83	82.71	1.231	0.067
Redness (a*)	12.21	14.29	16.04	1.083	0.367
Yellowness (b*)	3.50	2.08	3.58	0.446	0.318
Shear force (N)	26.61	27.16	27.83	0.533	0.666
Cooking loss (%)	15.62	16.67	16.40	0.622	0.787
Inosine monophosphate (ppm)	1196.04	1192.24	1193.84	22.224	0.998
Total cholesterol (μmol/mgprot)	54.34	69.41	45.66	5.309	0.185
**Thigh**					
pH	6.64	6.62	6.71	0.070	0.887
Lightness (L*)	77.58	79.79	81.17	1.131	0.445
Redness (a*)	14.92	15.62	15.00	0.357	0.695
Yellowness (b*)	4.29	4.38	3.83	0.479	0.892
Shear force (N)	26.15	26.07	26.38	0.625	0.981
Cooking loss (%)	15.32	13.89	14.51	0.588	0.630
Inosine monophosphate (ppm)	946.43	915.51	917.22	18.235	0.756
Total cholesterol (μmol/mgprot)	61.44	80.74	64.25	7.284	0.562

The data are the mean values obtained from eight replicates per treatment (two broilers per replicate). ^1^ SEM: standard error of the means.

## Data Availability

The data used and analyzed in the current study are available from the corresponding author on reasonable request.

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
