# Peer review of "Effects of Different Proportions of Amaranthus hypochondriacus Stem and Leaf Powder Inclusions on Growth Performance, Carcass Traits, and Blood Biochemical Parameters of Broilers"

_animals, 2023, doi:10.3390/ani13182818_

Round 1

Reviewer 1 Report

The proposed use of this Amaranthus hypochondriacus product is an interesting option to decrease the use of other common feed ingredients, but the paper needs to be deeply improved.

I am not qualified to assess the quality of English, but I could observe that this paper needs to be rewritten for several reasons, but one of them is that Extensive editing of English language is required.

Some methodological aspects required to understand the trial and to draw the conclusions are not described.

There are errors in the identification of some measured parameters, in the equations, in the tables descriptors, etc. (see notes in the attached file)

The description and discussion of results and  the conclusions should be improved.

A major revision of this paper is required.

Author Response

请参阅附件。

Reviewer 2 Report

The manuscript is written in poor quality, and there are a lot of grammatical and technical mistakes. The experimental treatments and replicates are not clear. Number of birds per treatment are not justifiable to validate the results. I am unable to understand why authors used RCBD.  In the formulation SO is varying at higher amount… how can authors state the results variation are due to AHSL or SO?. Overall discussion is without justification and results are not in line with justification. Authors fails to correlate different results. conclusion is not drawn on the basis of what are the results.

Line 12-16: authors did not carefully read the instruction of authors. Summary is not representing what it is meant for ‘Simple Summary: The use of Amaranthus hypochondriacus in poultry nutrition is not well docu- 12 mented. Hence this study will apply different proportions of Amaranthus hypochondriacus stem and 13 leaf powder (AHSL) as experimental diets to assess the nutritional value of broiler on growth per- 14 formance, apparent nutrient digestibility, carcass traits, meat quality, and blood biochemical param- 15 eters’

Try to provide summary keeping in view the author instructions

Line 19-20: why did you use ‘The experiment  applied a single-factor randomized block design’

Line 20-23: experimental treatments, replicates are not clear. Authors are advised to avoid wordy sentences. Please try to rewrite ‘8 replicate cages with 9 birds each were used in 20 each of the three dietary treatment groups, which included 216 male one-day-old broilers (Ross 308), 21 including a corn-soybean meal-based control diet and 2 experimental diets in which the control diet 22 was partly replaced by 3%, 6% (1-21 d) and 5%, 10% (22-42d) AHSL feed’

Line 23-25: where is p values ? ‘The results showed that 23 AHSL inclusion levels did not affect growth performance, carcass traits and meat quality on d 21 24 and 42’

Line 25-27: what do you mean by different abbreviation like DM and NDF ‘. However, AHSL inclusion levels decreased the apparent total tract nutrient digestibility 25 (ATND) of DM (p ˂ 0.001) and NDF (p ˂ 0.001); increased the serum concentration of phosphorus (p 26 ˂ 0.001) on d 21’ if you want to make it clear give full term on first seen like dry matter and neutral detergent fiber..

Line 28-30: what does abbreviation represent ALP and TP ‘Especially diet containing 10% AHSL significantly 28 increased the serum enzyme activity of ALP (p = 0.046) and the serum concentration of TP (p ˂ 0.001) 29 on d 42’

Line 30: how it can be inexpensive? ‘Overall, AHSL can be used as an inexpensive…..’ you did not give any result of profit or economic

Line 30-32: conclusion is not drawn on the basis of what are the results ‘Overall, AHSL can be used as an inexpensive and effective feed ingredient in broiler diets 30 and it can replace part of corn-soybean meal diet without any adverse effects, which is in favour of 31 saving feed resources and reducing breeding costs’

Introduction

Line 37-38: mistake in sentence. Please rewrite the sentence to clear the meaning of sentence ‘Approximately 60% to 80% of the overall cost of producing chicken is spent on feed, 37 making it the most expensive component. [1] ‘

Line 38-39: reference missing ‘Corn and soybean are two primary sources 38 of energy and protein in broiler feedstuff’

Line 39-41: rewrite the sentence, meanings are not clear ‘Considering the shortage of corn and soybean 39 and the cost of the feed formulation, grain amaranth might be used in place of some corn 40 or soybean meal in broiler diets because the price was more affordable,[2] ‘

Line 48-53: You are using leaf and stem powder and providing information ‘Amaranthus hypochondriacus has a high 48 nutritional value and roughly the same amount of gross energy as corn.[7] , the stems and 49 leaves of Amaranthus hypochondriacus have a very high crude protein content, which 50 ranges from 16 to 23 percent under dry matter conditions, and containing lysine, me- 51 thionine and other important amino acids, Lysine concentration is particularly high, at 52 around 1%[8] ‘ so what is the fun of providing the nutritional composition of grains ‘Furthermore, squalene had also been found in amaranth oil, which is an 53 important precursor of all steroids[9] . Squalene, fiber, and tocotrienols in amaranth grains 54 have been proven to lower blood serum cholesterol levels[10, 11] ‘

Line 55-57: this is not true for all species of livestock ‘Furthermore, anti-nutri- 55 tional factors (phytate, tannic acid, etc.) in raw amaranth also limited its use in livestock 56 and poultry[6

Line 58-65: keeping in view your statements ‘Previous research has revealed that adding untreated and heat-treated grain ama- 58 ranth meal to broiler diets in place of animal protein meat and bone meal has no discern- 59 ible impact on the broilers’ growth performance[12] . Fasuyi et al[13] reported that sun-dried 60 amaranthus cruentus leaf meal may be a source of rich nutrients for broilers and can be 61 included in the diets of broilers. When the added proportion in broilers' diets reaches 20%, 62 it has no effect on the health of broilers. Menyelo et al[14] found that there were not signif- 63 icant differences in the feed intake, body weight, and feed efficiency of broilers fed diets 64 containing 0%, 5%, 10%, 15%, and 20% amaranth cruentus leaf meal’ do you think your this sentence ‘But the use of Ama- 65 ranthus hypochondriacus in poultry nutrition is not well documented’ is correct?

Line 66-70:  you will apply or you had applied ‘Hence this study will 66 apply different proportions of Amaranthus hypochondriacus stem and leaf powder (AHSL) 67 as experimental diets to assess the nutritional value of broiler on growth performance, 68 apparent nutrient digestibility, carcass traits, meat quality, and blood biochemical param- 69 eters’

Line 90-91: why you used RCBD? ‘The experiment was carried out using a completely randomised block design of ex- 90 periments’

Line 91-94: ‘Totally,216 one-day-old, male broilers (Ross 308) were obtained from a local 91 hatchery (Zhengda Agriculture and Animal Husbandry Food Co., LTD. Xiangyang, 92 China). 8 replicate cages with 9 birds each were used in each of the three dietary treatment 93 groups’ not clear replication. Its number to small to validate the results

Line 100: what does ‘nutrient makeup’ represents

Table 2: what is AHSL? Give details in footnote

What does ‘TiO2’ represents ?

In the formulation SO is varying at higher amount… how can you state the results variation are due to AHSL or SO?

Line 124: apparent nutrient digestibility method is not correct especially for cp

Line 174-175: don’t abbreviate terms again and again ‘Results about body weight gain (BWG), average daily feed intake (ADFI), and feed 174 conversion ratio (FCR) on d 21 and d 42 of broilers were summarized in Table 3.’

Line 176-178: results are wrongly presented ‘However, the diets of 176 broilers having 3% and 6% AHSL inclusion levels had a significant tendency to increase 177 BWG compared to the control diet on d 21 (p < 0.1)’ check the table especially 22-42 days

Line 189: check ‘c’ of the ‘Control diet’

Line 188-191: if body weight is tending to increase and digestibility is decreasing how can you justify results ? ‘he ATND of DM and NDF in the diet with 5% and 10% AHSL decreased signifi- 188 cantly compared to the Control diet on d 42 (p = 0.025; p ˂ 0.001); and the ATND of CP 189 decreased significantly (p = 0.004); AHSL inclusion levels had no effect in the ATND of EE 190 on d 42 (p > 0.05). (Table 6)’

Table 4 missing foot note of abbreviation

Line 214-221: check the ‘c’ of control ‘There were no apparent differences (p > 0.05) in carcass traits between different ex- 214 perimental groups (Table 6), but the result showed that the value for all carcass traits was 215 higher in the diets with 5% and 10% AHSL inclusion levels compared to the Control diet. 216 Meat quality was no apparent difference (p > 0.05) detected both in the breast and 217 thigh (Table 7). However, the pH, redness of the breast and the lightness, redness of the 218 thigh in the diets with 5% and 10% AHSL inclusion levels were higher than the Control 219 diet; the lightness of the breast and the inosine monophosphate of the thigh was lower 220 than the Control diet’

Discussion is poorly written without justification. Some blood metabolites varying significantly but authors fail to correlate it

poor quality 

Reviewer 3 Report

The English could be improved. Some of the sentences were too long.

Author Response

请参阅附件。

Round 2

Reviewer 2 Report

Dear Authors

the manuscript is still not in readable form and the quality of the manuscript is still poor. I have provided you with a lot of suggestions to improve the manuscript technically and grammatically in the previous revision. however, there is still a lot of work to do to improve the manuscript technically and gramatically. 

A few serious concerns still need to be addressed 

Why did you use ‘The experiment applied a single factor randomized block design’?

Why do authors change percentage of AHSL in finisher phase? ‘The dietary treatments included a control diet, a 3% and 6% AHSL diet for 0-21 d, then the 3% and 6% AHSL diet were changed to 5% and 10% AHSL diet for 22-42 d

In the formulation SO is varying at higher amount… how can you state the results variation are due to AHSL or SO?

If body weight is tending to increase and digestibility is decreasing, how can you justify results ?

number of replicates are too small to validate the results.

digestibility methods??

Authors also fail to improve the discussion section.

conclusion is not drawn from the results. 

Serious concerns 
